# Sex-Specific Associations between Blood Pressure and Risk of Atrial Fibrillation Subtypes in the Tromsø Study

**DOI:** 10.3390/jcm10071514

**Published:** 2021-04-05

**Authors:** Hilde Espnes, Jocasta Ball, Maja-Lisa Løchen, Tom Wilsgaard, Inger Njølstad, Ellisiv B. Mathiesen, Eva Gerdts, Ekaterina Sharashova

**Affiliations:** 1Department of Community Medicine, UiT The Arctic University of Norway, 9019 Tromsø, Norway; maja-lisa.lochen@uit.no (M.-L.L.); tom.wilsgaard@uit.no (T.W.); inger.njolstad@uit.no (I.N.); ekaterina.e.sharashova@uit.no (E.S.); 2Centre for Research and Evaluation, Ambulance Victoria, Melbourne, VIC 3130, Australia; Jocasta.Ball@ambulance.vic.gov.au; 3Department of Epidemiology and Preventive Medicine, Monash University, Melbourne, VIC 3004, Australia; 4Baker Heart and Diabetes Institute, Melbourne, VIC 3004, Australia; 5Department of Clinical Medicine, Brain and Circulation Research Group, UiT The Arctic University of Norway, 9019 Tromsø, Norway; ellisiv.mathiesen@uit.no; 6Department of Neurology, University Hospital of North Norway, 9019 Tromsø, Norway; 7Center for Research on Cardiac Disease in Women, Department of Clinical Science, University of Bergen, 5020 Bergen, Norway; Eva.Gerdts@uib.no

**Keywords:** atrial fibrillation, blood pressure, hypertension, epidemiology, sex, longitudinal study

## Abstract

The aim of this study was to explore sex-specific associations between systolic blood pressure (SBP), hypertension, and the risk of incident atrial fibrillation (AF) subtypes, including paroxysmal, persistent, and permanent AF, in a general population. A total of 13,137 women and 11,667 men who participated in the fourth survey of the Tromsø Study (1994–1995) were followed up for incident AF until the end of 2016. Cox proportional hazards regression analysis was conducted using fractional polynomials for SBP to provide sex- and AF-subtype-specific hazard ratios (HRs) for SBP. An SBP of 120 mmHg was used as the reference. Models were adjusted for other cardiovascular risk factors. Over a mean follow-up of 17.6 ± 6.6 years, incident AF occurred in 914 (7.0%) women (501 with paroxysmal/persistent AF and 413 with permanent AF) and 1104 (9.5%) men (606 with paroxysmal/persistent AF and 498 with permanent AF). In women, an SBP of 180 mmHg was associated with an HR of 2.10 (95% confidence interval [CI] 1.60–2.76) for paroxysmal/persistent AF and an HR of 1.80 (95% CI 1.33–2.44) for permanent AF. In men, an SBP of 180 mmHg was associated with an HR of 1.90 (95% CI 1.46–2.46) for paroxysmal/persistent AF, while there was no association with the risk of permanent AF. In conclusion, increasing SBP was associated with an increased risk of both paroxysmal/persistent AF and permanent AF in women, but only paroxysmal/persistent AF in men. Our findings highlight the importance of sex-specific risk stratification and optimizing blood pressure management for the prevention of AF subtypes in clinical practice.

## 1. Introduction

Atrial fibrillation (AF) is the most common cardiac arrhythmia and is associated with severe complications such as stroke, heart failure, premature death, and reduced quality of life [1]. Affecting over 33 million people worldwide [2] and 15–20% of women and men between 70 and 79 years of age [3], the condition has a significant social and economic impact. The prevalence of AF is increasing in parallel to population aging and to increases in chronic disease epidemics related to unhealthy lifestyle [4,5]. It has been demonstrated that more than half of the AF burden is potentially preventable through management of classic cardiovascular risk factors [6]. One of the most important contributors to the AF burden is hypertension, which accounts for one-fifth of AF cases [6]. It has also been shown that tight blood pressure control may reduce the risk of AF up to 50% [7]. Hypertension promotes electrical and structural remodeling of the left atria, which predisposes to AF and is associated with a 1.8-fold increased risk of incident AF [8]. In addition, previous studies have shown that hypertension promotes the progression from paroxysmal AF to more sustained forms of the disease [8,9,10]. Many of these studies investigated the effect of hypertension in patients with established paroxysmal AF, while longitudinal studies examining the association between blood pressure and AF subtypes in a population without AF at baseline are sparse [11,12]. 

Sex differences in the epidemiology of AF are well known [1,13,14] and may also exist in the association between blood pressure and AF subtypes. The incidence of AF is almost twice as high among men compared to women [13], and sex differences in the prevalence of AF risk factors such as hypertension, increased body mass index (BMI), diabetes mellitus, and coronary heart disease are well described [14]. The Tromsø Study demonstrated that long-term blood pressure trajectories were more strongly associated with the risk of AF in women than men [15]. However, less is known about the impact of blood pressure on the sex-specific risk of different AF subtypes. Although both elevated systolic blood pressure (SBP) and diastolic blood pressure (DBP) have been associated with an increased risk of AF, SBP has been shown to be a superior predictor of incident AF [16,17,18]. We therefore aimed to explore sex-specific associations between SBP, hypertension, and the risk of incident AF subtypes, including paroxysmal, persistent, and permanent AF, using data from the population-based Tromsø Study.

## 2. Materials and Methods

### 2.1. Study Design and Participants

The Tromsø Study is a longitudinal cohort study of the general population in the municipality of Tromsø, Northern Norway [19,20]. In brief, the study was initiated in 1974 to identify cardiovascular risk factors that could be modified through population strategies for the primary prevention of cardiovascular disease. Seven consecutive surveys have been conducted between 1974 and 2016, to which total birth cohorts and random population samples were invited. The present analysis included participants from the fourth survey (Tromsø 4) conducted in 1994–1995. A total of 27,158 persons aged 25–97 years attended (77% of those invited). Of these, 26,992 participants provided informed consent and were eligible for inclusion in the present analyses. Individuals with missing blood pressure data, insufficient information for AF validation, or a history of AF were excluded (Figure 1), leaving a final study population of 24,804 participants (13,137 women and 11,667 men). 

The Tromsø Study has been approved by the Regional Committee for Medical and Health Research Ethics and by the Norwegian Data Protection Authority and was performed in accordance with the Declaration of Helsinki. All participants included in the present study provided written informed consent and could withdraw their consent at any time. 

### 2.2. Data Collection

Following a standardized protocol, data were collected via questionnaires, physical examinations, and blood samples. During the physical examination, SBP and DBP were measured using an oscillometric digital automatic device (Dinamap Vital Signs Monitor 1846; Critikon Inc., Tampa, FL, USA) and an appropriately sized cuff for the upper right arm of the individual participant. Blood pressure was recorded by a trained nurse after 2 min of rest in a seated position and was measured three times in total with 1 min intervals. The mean of the last two measurements was used in the analyses. Hypertension was defined as SBP ≥ 140 mmHg and/or DBP ≥ 90 mmHg, and/or current antihypertensive medication use. Blood samples were analyzed by the Department of Laboratory Medicine, University Hospital of North Norway (Tromsø), to determine non-fasting serum concentrations of total cholesterol, high-density lipoprotein (HDL) cholesterol, and triglycerides. Weight and height were used to calculate BMI (weight (kg) divided by height (m) squared). Information on the current use of antihypertensive medications (yes/no), current daily smoking (yes/no), education (primary or partly secondary education/upper secondary education/college or university less than 4 years/college or university 4 years or more), and history of myocardial infarction, angina pectoris, stroke, and diabetes mellitus (yes/no) was collected from the questionnaire. Leisure time physical activity levels were categorized into four groups (inactive/low activity/moderate activity/high activity) by combining two questions on the number of hours per week of light and hard activity [21]. 

### 2.3. Follow-Up and Detection of Incident Atrial Fibrillation

The follow-up period for each individual participant began at the date of the physical examination and ended at the date of first documented AF, emigration or death (identified through the Population Register of Norway), or the end of the follow-up period (31 December 2016), whichever came first. Incident cases of AF were detected through linkage to the diagnosis registry of the University Hospital of North Norway, the only hospital in this area, by using the unique Norwegian national 11-digit identification number. The registry includes diagnoses from both out- and in-patient clinics, and incident cases of AF were detected by manual and/or electronic text searches in paper (used until 2001) and digital versions of hospital records using the International Classification of Diseases, 9th Revision (ICD–9) codes 427.0 to 427.99 and the International Classification of Diseases, 10th Revision (ICD–10) codes I47 and I48. In addition, for participants with a diagnosis of a cardiovascular event (ICD–9 codes 410–414, 428, 430–438, 798–799, and/or ICD–10 codes I20–I25, I50, I60–69, R96, R98, and R99) but no recorded diagnostic code for arrhythmia, hospital records were searched for AF. The diagnosis was confirmed by AF documented on an electrocardiogram and validated by an independent endpoint committee following a detailed protocol [19]. 

Classification of AF subtype was performed according to the 2016 European Society of Cardiology guidelines for the management of AF: paroxysmal (self-terminating and lasting ≤7 days); persistent (lasting >7 days, including episodes requiring intervention to terminate); permanent (sustained AF where repeated attempts to restore sinus rhythm have failed or were not attempted and the AF rhythm is accepted) [22]. The latest AF subtype registered in the hospital record was used. Since the distinction between paroxysmal and persistent AF often requires long-term monitoring [23], which was usually not possible for the endpoint committee to obtain, these two subtypes were combined. Postoperative AF occurring within 28 days after surgery, AF related to myocardial infarction within 28 days of the event, AF related to other acute cardiac events or within 28 days after an event, as well as AF occurring during the last 7 days of life, were all classified as non-cases. 

### 2.4. Statistical Analyses

Characteristics of the study population are presented as means (standard deviations) for continuous variables and as numbers (percentages) for categorical variables, stratified by sex. Adjustment for age and comparisons between participants with no AF and those with the respective subtypes (paroxysmal/persistent and permanent) were conducted using linear regression for continuous variables and logistic regression for categorical variables and estimated for a mean age of 46 years.

In the primary analysis, Cox proportional hazards regression models were used to estimate the association between SBP and AF subtypes. Fractional polynomials modeling was used to reveal non-linear associations. Hazard ratios (HRs) and 95% confidence intervals (CIs) were calculated using AF subtype as the dependent variable and fractional polynomials of SBP as the main exposure using an SBP of 120 mmHg as the reference value. In addition, the models were adjusted for fractional polynomials of age and for BMI, total cholesterol, current smoking, leisure time physical activity, and history of myocardial infarction, angina pectoris, stroke, and diabetes mellitus. 

In a secondary analysis, Cox regression models (adjusted for the same covariates as in the primary analysis) were also used to assess the association between hypertension groups and the risk of AF subtypes. Normotensive participants (SBP < 140 mmHg and DBP < 90 mmHg and no current antihypertensive medication use) were used as the reference and compared to participants with controlled hypertension (SBP < 140 mmHg and DBP < 90 mmHg and current antihypertensive medication use), uncontrolled hypertension (SBP ≥ 140 mmHg and/or DBP ≥ 90 mmHg and current antihypertensive medication use), and untreated hypertension (SBP ≥ 140 mmHg and/or DBP ≥ 90 mmHg and no current antihypertensive medication use).

We also performed a sensitivity analysis, which was restricted to participants without history of myocardial infarction, angina pectoris, stroke, and diabetes mellitus, as they could potentially act as moderators for the association between SBP and the risk of incident AF subtypes through pathophysiologic mechanisms that predispose to AF. We performed an additional sensitivity analysis where only those with a history of coronary heart disease (myocardial infarction and angina pectoris) were excluded. Tests for interaction between SBP and sex were performed by including the cross-product term in the models. The same interaction analysis was performed to test for the interaction between hypertension groups and sex. All statistical analyses were sex-specific and performed using SAS 9.4 (SAS institute, Cary, NC, USA). The proportional hazard assumption was assessed with graphical inspection of log minus log survival curves between quartiles of continuous variables or between categories of nominal variables. A two-sided *p*-value < 0.05 was considered statistically significant. 

## 3. Results

### 3.1. Baseline Characteristic 

Women who developed permanent AF during follow-up were older and had higher mean SBP at baseline compared to women who developed paroxysmal/persistent AF (Table 1). In addition, women who developed AF had significantly higher BMI at baseline compared to women who did not develop AF, with a proportional linear relationship to increasing severity of AF. The proportion of women with hypertension and antihypertensive medication use was slightly higher among those who developed permanent AF compared to women who developed paroxysmal/persistent AF. Men who developed permanent AF during follow-up were older, had slightly lower SBP and DBP and marginally higher BMI at baseline than men who developed paroxysmal/persistent AF. The proportion of men with hypertension and antihypertensive medication use at baseline was similar between AF subtypes. Both women and men who developed AF had more comorbidities than those who did not develop AF. Among women who developed paroxysmal/persistent AF, the proportion who reported a history of angina pectoris and diabetes mellitus at baseline was higher than that among women who developed permanent AF. The proportion of men reporting a history of angina pectoris and diabetes mellitus at baseline was higher among those who developed paroxysmal/persistent AF, while men who developed permanent AF more often reported a history of stroke. 

### 3.2. Systolic Blood Pressure and Incident Atrial Fibrillation Subtypes

Over a mean follow-up of 17.6 ± 6.6 years, incident AF occurred in 914 women (7.0%) and 1104 men (9.5%), of whom 501 and 606, respectively, developed paroxysmal/persistent AF. Among women, SBP was significantly associated with the risk of both paroxysmal/persistent and permanent AF (Figure 2A,B). The association was strongest for paroxysmal/persistent AF, where an SBP of 150 mmHg was associated with an HR of 1.62 (95% CI 1.36–1.93), and an SBP of 180 mmHg was associated with an approximately 2-fold higher risk of incident AF as compared to the reference SBP value of 120 mmHg (HR 2.10, 95% CI 1.60–2.76). For permanent AF in women, the corresponding HRs were 1.46 (95% CI 1.20–1.78) and 1.80 (95% CI 1.33–2.44). 

In men, SBP was linearly associated with the risk of paroxysmal/persistent AF (Figure 2C). Compared to men with an SBP of 120 mmHg, men with an SBP of 150 mmHg had a 38% increased risk of incident paroxysmal/persistent AF (HR 1.38, 95% CI 1.21–1.57), while men with an SBP of 180 mmHg had almost twice the risk of incident paroxysmal/persistent AF (HR 1.90, 95% CI 1.46–2.46). We did not observe a significant association between SBP and the risk of permanent AF in men (Figure 2D). There was a significant interaction between SBP and sex (*p* < 0.001) for permanent AF, but not for paroxysmal/persistent AF (*p* = 0.156). 

In both sensitivity analyses excluding participants with a history of comorbidities, the associations between SBP and AF subtypes remained similar, except for permanent AF in men, where a significant association with SBP was identified, similar to that found in women (Appendix A). Interaction between SBP and sex for permanent AF remained significant (*p* = 0.0048 and *p* = 0.0017, respectively). 

### 3.3. Hypertension and Incident Atrial Fibrillation Subtypes

Compared to the normotensive group, women with controlled hypertension had no increased risk of incident paroxysmal/persistent AF, while an increased risk (HR 3.22, 95% CI 1.87–5.55) was seen for permanent AF (Table 2). Uncontrolled hypertension was associated with an increased risk of both paroxysmal/persistent AF (HR 2.53, 95% CI 1.85–3.46) and permanent AF in women (HR 3.11, 95% CI 2.23–4.35). The same pattern was observed for untreated hypertension, with HRs of 1.70 (95% CI 1.35–2.15) and 1.58 (95% CI 1.19–2.09), respectively.

In men, controlled hypertension was not associated with an increased risk of paroxysmal/persistent AF, while a significantly increased risk of permanent AF was found (HR 1.82, 95% CI 1.11–2.98) (Table 2). Men with uncontrolled hypertension had an increased risk of both paroxysmal/persistent and permanent AF (HR 1.89, 95% CI 1.41–2.52 and HR 1.66, 95% CI 1.22–2.27, respectively), while men with untreated hypertension only had an increased risk of paroxysmal/persistent AF (HR 1.24, 95% CI 1.03–1.49). There was a significant interaction between uncontrolled and untreated hypertension and sex for paroxysmal/persistent AF (*p* = 0.0145 and *p* < 0.001, respectively) and for untreated hypertension and sex for permanent AF (*p* < 0.001). Age-adjusted means for SBP in each hypertension group are presented in Appendix A.

## 4. Discussion

Sex differences in the association between SBP and the risk of AF have been demonstrated previously [15]. The present study adds to this knowledge by showing that the associations between SBP and hypertension and the risk of AF subtypes are sex-specific. We found that increasing SBP was associated with a higher risk of both AF subtypes. For paroxysmal/persistent AF, the association with SBP was similar in both sexes, while for permanent AF, the association was found in women only. Uncontrolled hypertension increased the risk of both paroxysmal/persistent and permanent AF in both sexes. The same was seen for untreated hypertension, except for permanent AF in men, where no increased risk was observed compared to normotensive participants. The risk of permanent AF was also elevated in women and men with controlled hypertension. All associations were stronger in women compared to men.

Previous studies have focused on SBP as a risk factor for progression of paroxysmal AF to more sustained forms of the disease [9,10]. Our study adds to this by showing the risk of developing different subtypes of AF in a population without AF at baseline. We found that the association between SBP and incident AF was strongest for paroxysmal/persistent AF. A case–control study conducted by Thomas et al. on patients taking antihypertensive treatment found that the association between average achieved SBP and incident AF was stronger for persistent/intermittent AF and sustained AF than transitory AF [11]. In the same study, they also observed a J-shaped relationship between SBP and AF, with an increasing risk of AF in those with SBP < 120 mmHg. The latter contrasts with our findings, which showed that SBP values < 120 mmHg were associated with the lowest risk of incident AF in all groups, except for the association with permanent AF in men, which was moderated by coronary heart disease. However, in the study by Thomas et al., all participants had treated hypertension, and therefore represent a different group than our sample from the general population. In our study, participants with an SBP < 120 mmHg predominantly had no history of hypertension. Our results are further supported by the findings of Verdecchia et al. [7], which found that tight blood pressure control (SBP < 130 mmHg) was associated with a 54% reduced risk of new-onset AF compared to usual blood pressure control (SBP < 140 mmHg). It has also been shown that pre-hypertension (blood pressure between 120–139/80–89 mmHg and no antihypertensive medication use) is associated with a higher risk of AF [24], which is consistent with our finding of an increased risk of AF for an SBP of 130 mmHg compared to an SBP of 120 mmHg.

In a study examining the risk of incident AF in relation to on-treatment SBP of hypertensive patients with electrocardiographic signs of left ventricular hypertrophy (LVH), both patients with on-treatment SBP between 131–141 mmHg and patients with on-treatment SBP ≤ 130 mmHg had a lower risk of new-onset AF compared to patients with on-treatment SBP ≥ 142 mmHg [25]. By comparing hypertension groups, we found that uncontrolled hypertension was associated with a higher risk of both AF subtypes. Of note, we found that individuals with controlled hypertension had a risk of paroxysmal/persistent AF that was comparable to that of normotensive participants, while controlled hypertension was associated with an increased risk of permanent AF in both sexes, but especially in women. It has been well demonstrated that high levels of SBP predispose to LVH and to dilatation of the left atria, which further increases the risk of AF [8,26,27,28]. Therefore, previously hypertensive participants in our study population may still be predisposed to AF, even if their blood pressure was controlled at the time of the physical examination.

Sex differences in many aspects of AF are well established [29], from occurrence, cardiometabolic risk factors, preclinical disease and clinical presentation to treatment and outcomes [30,31,32]. Some previous studies exploring the sex-specific association between SBP and AF have demonstrated similar strengths of associations in women and men [33,34], while others have found a stronger association in women [15]. However, studies describing this association in AF subtypes are sparse. Therefore, the present results add to current knowledge by demonstrating that elevated SBP seems to play a more important role in the development of AF subtypes in women than in men. We found that women in all categories of SBP and hypertension groups had a higher risk of incident paroxysmal/persistent and permanent AF than men. Women with hypertension have been shown to have a higher prevalence of subclinical cardiac damage, such as LVH [35], and although women in general have a lower cardiovascular risk than men, this seems to be offset in hypertensive women with LVH [36]. In addition, despite aggressive antihypertensive treatment, LVH is less likely to be reversed in women than men [35]. This could explain why we found higher risk estimates for the associations between all categories of SBP and hypertension groups and incident AF for women than for their male counterparts.

However, a major concern is that the sex-specific findings in our study might have been moderated by the difference in the prevalence of coronary heart disease. Coronary heart disease increases the risk of AF [37]. Previous research has shown that coronary heart disease is more common in men than women with AF [1], which is consistent with what we found in our study. After the exclusion of participants with myocardial infarction and angina pectoris, the associations between SBP and permanent AF in men became significant and more similar to the those in women, but the interaction between SBP and sex remained significant.

Our findings have several clinical implications and highlight the value of early detection and treatment of hypertension to prevent AF, as well as predisposing conditions such as LVH and dilatation of the left atria. It has been shown that lowering SBP is beneficial for prevention of AF [25,38], which is consistent with our findings. The risk of incident AF, especially paroxysmal/persistent AF, decreased with lower SBP, which implicates that antihypertensive treatment could be more beneficial in the prevention of paroxysmal/persistent AF. The fact that controlled hypertension is associated with an increased risk of permanent AF further emphasizes the importance of the primary prevention of hypertension. We found stronger associations between SBP and AF subtypes, especially permanent AF, in women compared to men, which may demonstrate the necessity of sex-specific optimal blood pressure targets and/or treatment thresholds, as well as sex-specific AF risk stratification.

This study has several strengths, including the prospective design, the large number of participants, the high attendance rate, and the long-term follow-up, which provides a solid base for the ascertainment of the sex-specific association between SBP and incident AF subtypes. Our data also included a broad diversity of variables regarding health, lifestyle, and physical measurements, which allowed us to adjust for multiple potential confounders. All AF cases were confirmed and validated according to current guidelines by an independent endpoint committee.

However, there are study limitations that require comment. First, some variables were self-reported, which could lead to over- or underestimation of the prevalence of risk factors. Second, changes in blood pressure, antihypertensive medication use, and occurrence of comorbidities during follow-up may have influenced the risk of incident AF. For example, if a participant started to use antihypertensive medication after inclusion in the fourth survey, this would change their SBP category, which in turn could affect the association between SBP and the AF subtypes in our analyses. However, updating baseline characteristics in a time-dependent analysis for those participants who also attended one or more of the following Tromsø Study surveys had virtually no effect on the associations (data not presented). Furthermore, long-term patterns of SBP (trajectories) [15] could provide additional prognostic information for the prediction of AF subtypes, but these analyses would require a larger amount of validated AF cases to ensure sufficient statistical power. Third, in our study, we had no information on vascular stiffness, left ventricular diastolic dysfunction, chronic pulmonary disorders, rheumatic disease, hypertensive pregnancy disorders, and menopausal status, which could have interfered with the results. Fourth, ambulatory blood pressure monitoring (ABPM) or home blood pressure monitoring was not used in this study; thus, white-coat hypertension could not be detected [39]. Even though research recordings seem to be closer to ABMP than measurements from routine clinical practice [40], it is still likely that some cases of masked or white-coat hypertension were misclassified in the present study. Fifth, although all AF cases were validated by skilled physicians in the independent end-point committee, there is still a potential risk of misclassification of AF subtypes. Furthermore, it is likely that some participants have undiagnosed AF, due to silent AF and/or paroxysmal AF that fails to be detected on examination, and because some AF patients are treated by their general practitioner and are not referred to the hospital. The total AF prevalence may therefore be higher than what is presented here. Sixth, to avoid potential misclassification bias, we have excluded participants with insufficient data on AF (Figure 1). Investigating their baseline characteristics, we found that they had a comparable risk profile to the participants with AF and including these as AF cases would therefore probably further strengthen our results.

## 5. Conclusions

In this large population-based cohort study, elevated SBP was associated with an increased risk of both paroxysmal/persistent AF and permanent AF in women, and with paroxysmal/persistent AF in men. Uncontrolled hypertension increased the risk of both paroxysmal/persistent and permanent AF in both sexes, while untreated hypertension increased the risk of both subtypes in women, but only paroxysmal/persistent AF in men. Risk of permanent AF was also elevated in women and men with controlled hypertension. These associations were stronger in women compared to men. Our findings carry important implications for public health, as they highlight the importance of timely preventive measures and optimizing blood pressure management. By addressing sex-specific differences in the association between SBP and the risk of AF subtypes, we provide further knowledge in the field of sex-specific risk stratification and prevention of AF subtypes in clinical practice.

## Figures and Tables

**Figure 1 jcm-10-01514-f001:**
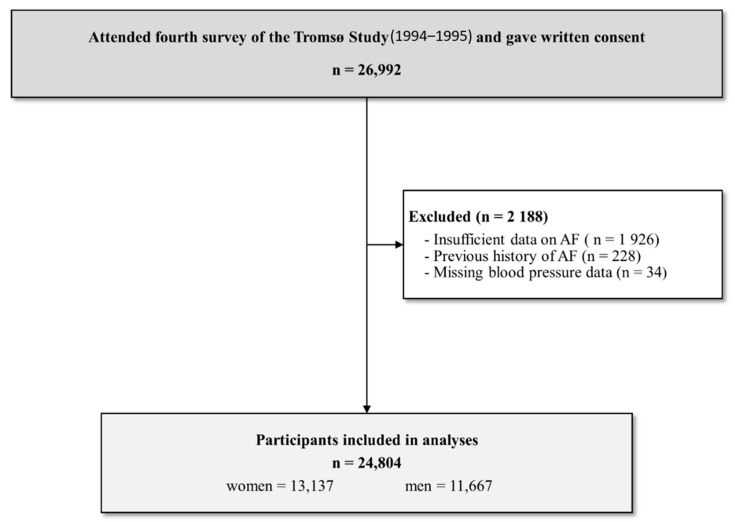
Flow chart of study participants in the fourth survey of the Tromsø Study. AF indicates atrial fibrillation.

**Figure 2 jcm-10-01514-f002:**
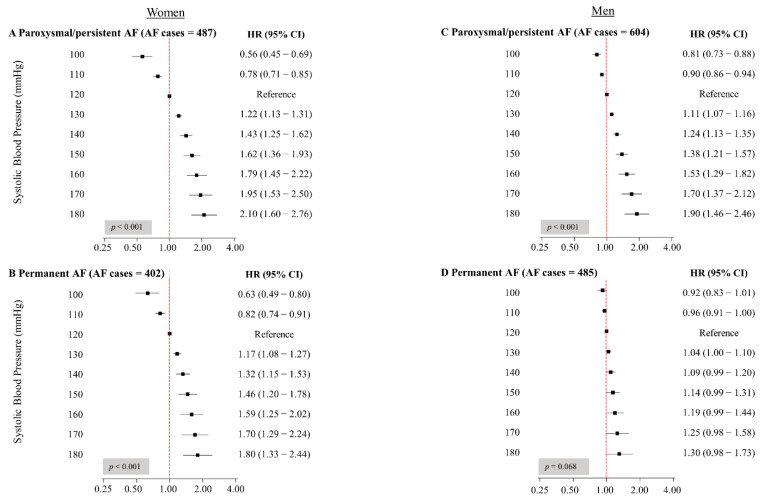
Hazard ratios for the associations between systolic blood pressure and AF subtypes in women (**A**,**B**) and men (**C**,**D**) (The Tromsø Study). AF indicates atrial fibrillation; CI, confidence interval; HR, hazard ratio. HRs and 95% CIs were calculated using AF subtype as the dependent variable and fractional polynomials of SBP as the exposure using SBP of 120 mmHg as the reference value. All HRs are adjusted for age, body mass index, total cholesterol, current smoking, leisure time physical activity, and history of myocardial infarction, angina pectoris, stroke, and diabetes mellitus. Due to missing observations on covariates, numbers for AF cases are marginally less.

**Table 1 jcm-10-01514-t001:** Baseline characteristics according to AF subtype during follow-up (The Tromsø Study).

	Women				Men			
Characteristics	No AF	Parox/Pers	Perm	*p*-Value	No AF	Parox/Pers	Perm	*p*-Value
*n* (%)	12.223 (93.0)	501 (3.8)	413 (3.1)		10.563 (90.5)	606 (5.2)	498 (4.3)	
Age, years	44.4 (14.0)	62.3 (11.9)	67.3 (9.2)	<0.001	43.8 (13.0)	56.7 (12.7)	60.7 (10.8)	<0.001
Systolic blood pressure, mmHg	130.3 (20.2)	137.1 (25.6)	138.1 (24.0)	<0.001	136.3 (15.8)	140.7 (21.4)	139.3 (21.5)	<0.001
Diastolic blood pressure, mmHg	75.7 (11.9)	78.3 (14.2)	77.7 (13.5)	<0.001	79.4 (11.1)	81.3 (12.4)	81.0 (13.1)	<0.001
Hypertension ^a^, n (%)	3051 (21.4)	340 (33.1)	326 (34.9)	<0.001	3879 (38.4)	364 (47.7)	325 (48.5)	<0.001
Antihypertensive medication use, *n* (%)	481 (2.6)	94 (5.2)	115 (6.7)	<0.001	413 (2.9)	90 (5.6)	88 (5.7)	<0.001
Body mass index, kg/m^2^	24.6 (4.1)	25.6 (4.7)	26.6 (5.4)	<0.001	25.5 (3.2)	26.1 (3.6)	26.9 (3.5)	<0.001
Total cholesterol, mmol/L	6.00 (1.35)	6.06 (1.34)	5.84 (1.22)	0.014	6.02 (1.21)	6.00 (1.14)	5.97 (1.13)	0.677
HDL cholesterol, mmol/L	1.64 (0.40)	1.61 (0.41)	1.60 (0.48)	0.043	1.35 (0.35)	1.31 (0.35)	1.32 (0.37)	0.023
Triglycerides, mmol/L	1.31 (0.81)	1.45 (1.18)	1.45 (1.08)	<0.001	1.77 (1.19)	1.77 (0.99)	1.82 (1.15)	0.726
Current daily smoking, *n* (%)	4612 (36.9)	159 (38.4)	79 (25.8)	<0.001	4070 (38.2)	174 (30.1)	144 (30.9)	<0.001
Leisure time physical activity n (%)								
Inactive	1020 (7.6)	76 (7.2)	76 (7.5)	0.875	822 (7.9)	51 (6.4)	47 (6.8)	0.221
Low activity	5201 (43.0)	214 (43.5)	176 (43.2)	0.977	4168 (39.7)	223 (37.4)	174 (36.2)	0.212
Moderate activity	5367 (43.9)	184 (41.8)	146 (41.6)	0.526	4472 (42.9)	286 (46.6)	233 (46.5)	0.088
High activity	505 (3.7)	20 (6.3)	11 (5.1)	0.046	997 (8.7)	45 (9.1)	37 (10.3)	0.602
Education, n (%)								
Primary/partly secondary education	4097 (33.8)	300 (33.6)	296 (38.7)	0.207	2963 (28.8)	253 (30.0)	202 (25.3)	0.145
Upper secondary education	4388 (34.6)	126 (33.7)	67 (25.2)	0.005	4028 (37.3)	192 (35.5)	181 (42.0)	0.079
College/university less than 4 years	1832 (12.8)	37 (12.8)	23 (12.4)	0.992	1791 (16.1)	73 (14.2)	66 (16.7)	0.465
College/university 4 years or more	1857 (13.1)	34 (11.4)	22 (11.4)	0.563	1756 (15.3)	85 (17.4)	47 (13.1)	0.223
History of myocardial infarction, *n* (%)	97 (0.3)	21 (0.6)	18 (0.5)	0.085	251 (1.4)	53 (2.3)	52 (2.2)	0.001
History of angina pectoris, *n* (%)	249 (0.7)	58 (1.3)	51 (1.0)	<0.001	286 (1.5)	84 (3.6)	67 (2.7)	<0.001
History of stroke, *n* (%)	99 (0.5)	18 (1.0)	9 (0.4)	0.090	110 (0.7)	9 (0.4)	23 (1.2)	0.028
History of diabetes mellitus, *n* (%)	128 (0.8)	28 (1.8)	19 (1.2)	<0.001	113 (0.8)	24 (1.4)	13 (0.8)	0.045

AF indicates atrial fibrillation; HDL, high-density lipoprotein; parox, paroxysmal AF; pers, persistent AF; perm, permanent AF. Values are mean (SD) or number (%); the means (except age means) and percentages are adjusted for age using linear or logistic regression models, respectively, and estimated for a mean age of 46 years. ^a^ Hypertension was defined as systolic blood pressure ≥140 mmHg and/or diastolic blood pressure ≥90 mmHg and/or current use of antihypertensive medications.

**Table 2 jcm-10-01514-t002:** Hazard ratios for the association between hypertension groups and atrial fibrillation subtypes in women and men (The Tromsø Study).

	Women				Men			
	Paroxysmal/Persistent AF		Permanent AF		Paroxysmal/Persistent AF		Permanent AF	
HT Group ^a^	HR (95% CI)	*p*-Value	HR (95% CI)	*p*-Value	HR (95% CI)	*p*-Value	HR (95% CI)	*p*-Value
Normotensive	Reference		Reference		Reference		Reference	
Controlled HT	1.71 (0.92−3.16)	0.088	3.22 (1.87−5.55)	<0.001	1.24 (0.73−2.09)	0.426	1.82 (1.11−2.98)	0.018
Uncontrolled HT	2.53 (1.85−3.46)	<0.001	3.11 (2.23−4.35)	<0.001	1.89 (1.41−2.52)	<0.001	1.66 (1.22−2.27)	0.001
Untreated HT	1.70 (1.35−2.15)	<0.001	1.58 (1.19−2.09)	0.002	1.24 (1.03−1.49)	0.022	1.15 (0.93−1.42)	0.195

AF indicates atrial fibrillation; HR, hazard ratio; CI, confidence interval; HT, hypertension. HRs are adjusted for age, body mass index, total cholesterol, current smoking, leisure time physical activity, and comorbidities (history of myocardial infarction, angina pectoris, stroke, and diabetes mellitus). ^a^ Normotensive: SBP < 140 mmHg and DBP < 90 mmHg and no current antihypertensive medication use. Controlled hypertension: SBP < 140 mmHg and DBP < 90 mmHg and current antihypertensive medication use. Uncontrolled hypertension: SBP ≥ 140 mmHg and/or DBP ≥ 90 mmHg and current antihypertensive medication use. Untreated hypertension: SBP ≥ 140 mmHg and/or DBP ≥ 90 mmHg and no current antihypertensive medication use.

## Data Availability

The data underlying this article were provided by Tromsøundersøkelsen by permission. The data are available upon reasonable request and application for data access to the Tromsø Study. More information may be found on http://www.tromsoundersokelsen.no (accessed on 11 November 2019).

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
