# Peer review of "Sex-Specific Associations between Blood Pressure and Risk of Atrial Fibrillation Subtypes in the Tromsø Study"

_jcm, 2021, doi:10.3390/jcm10071514_

Round 1
Reviewer 1 Report
This is a very interesting study investigating sex-specific associations between blood pressure and risk of atrial fibrillation subtypes in a population-based study. The manuscript is well-written. Study findings are novel based on appropriate statistical methods of data analyses. Please find below some other suggestions or comments for the manuscript.
- Missing data were excluded in the analyses. Would you please comment on are there any potential risks of selection bias? Whether the differences in characteristics between those included and those excluded in the analyses.
- Please provide STROBE checklist for this study. Of note that curtailing missing bias, as an important component in the STROBE checklist, need to be performed for cohort study.
- How many cases included in the sensitivity analyses excluding people who reported MI, angina pectoris, stroke and diabetes?
- Sex differences in the association between SBP and AF incidence was reported in the same Tromsø study (ref 16). The previous study findings were based on analyses of long-term exposure to risk factors which improve the prediction of AF compared to single time point measurements as presented in the current manuscript. Acknowledging the novel findings on the role of SBP on two main types of AF, the limitation of the use of mean values of the last two BP measurements need to be noted in the discussion.
- The potential effect modification of CHD on the sex-specific findings in the study is acknowledge in the discussion section. The authors mentioned that after exclusion of people with MI and angina pectoris, the associations between SBO and permanent AF in men became more similar to women, although interaction with sex remained significant. The data were not presented but I think including the sensitivity analyses (and acknowledge the methods somewhere) in the supplements would be very good.
Reviewer 2 Report
This study describes a large dataset of male and female patients from the Tromso study. It was found that sex differences in the association between SBP values and AF do exist. The study is well done and written. I have only a few remarks:
- Is is known whether the duration of elevated SBP or poor control during a longer period of time is associated with the pattern of AF? Is that different among men and women?
- women more often have LVH compared to men as you state in the discussion, do you have any data on vascular stiffness and signs of diastolic disfunction in this population that may have interfered with the results?
- do you have information on non-cardiac co-morbidities that may have interfered with the presence of AF, such as chronic pulmonary disorders or rheumatic disease (with lower Hb than usual)?
- Do you have sex-specific parameters in this cohort, such as related to hypertensive pregnancy disorders and menopausal status?
Round 2
Reviewer 1 Report
Thank you for the revision. All my comments are addressed accordingly.